# Axonal Injuries Cast Long Shadows: Long Term Glial Activation in Injured and Contralateral Retinas after Unilateral Axotomy

**DOI:** 10.3390/ijms22168517

**Published:** 2021-08-07

**Authors:** María José González-Riquelme, Caridad Galindo-Romero, Fernando Lucas-Ruiz, Marina Martínez-Carmona, Kristy T. Rodríguez-Ramírez, José María Cabrera-Maqueda, María Norte-Muñoz, Manuel Vidal-Sanz, Marta Agudo-Barriuso

**Affiliations:** 1Grupo de Oftalmología Experimental, Instituto Murciano de Investigación Biosanitaria Virgen de la Arrixaca (IMIB-Arrixaca), 30120 Murcia, Spain; mariajosefa.gonzalez3@um.es (M.J.G.-R.); fer138lucas@gmail.com (F.L.-R.); marina.m.c1@um.es (M.M.-C.); kristytatiana.rodriguez@um.es (K.T.R.-R.); josemaria.olvera@gmail.com (J.M.C.-M.); maria.norte@um.es (M.N.-M.); manuel.vidal@um.es (M.V.-S.); 2Departamento de Oftalmología, Facultad de Medicina, Universidad de Murcia, 30100 Murcia, Spain; 3Center of Neuroimmunology and Department of Neurology, Hospital Clinic of Barcelona, Institut d’Investigacions Biomèdiques August Pi Sunyer (IDIBAPS), University of Barcelona, 08036 Barcelona, Spain

**Keywords:** M1, M2, microglial cells, optic nerve crush, retina, inflammation, bilateral response, astrocytes, Müller cells

## Abstract

Background: To analyze the course of microglial and macroglial activation in injured and contralateral retinas after unilateral optic nerve crush (ONC). Methods: The left optic nerve of adult pigmented C57Bl/6 female mice was intraorbitally crushed and injured, and contralateral retinas were analyzed from 1 to 45 days post-lesion (dpl) in cross-sections and flat mounts. As controls, intact retinas were studied. Iba1^+^ microglial cells (MCs), activated phagocytic CD68^+^MCs and M2 CD206^+^MCs were quantified. Macroglial cell changes were analyzed by GFAP and vimentin signal intensity. Results: After ONC, MC density increased significantly from 5 to 21 dpl in the inner layers of injured retinas, remaining within intact values in the contralateral ones. However, in both retinas there was a significant and long-lasting increase of CD68^+^MCs. Constitutive CD206^+^MCs were rare and mostly found in the ciliary body and around the optic-nerve head. While in the injured retinas their number increased in the retina and ciliary body, in the contralateral retinas decreased. Astrocytes and Müller cells transiently hypertrophied in the injured retinas and to a lesser extent in the contralateral ones. Conclusions: Unilateral ONC triggers a bilateral and persistent activation of MCs and an opposed response of M2 MCs between both retinas. Macroglial hypertrophy is transient.

## 1. Introduction

The retina is part of the central nervous system (CNS) and is a widely used model to study the response of CNS neurons and glial cells to insults and potential neuroprotective therapies.

Glial cells are pivotal in the maintenance of CNS homeostasis and neuronal well-being, so glial deregulation leads to neuronal disfunction and death [1,2,3]. In the retina, the three major glial cells are microglial cells (MCs) [4], astrocytes and Müller cells, the latter a radial glia unique to the retina [5]. Each type has its own specific function and role, but they all activate in response to stress, injury and neuronal death to minimize the damage and restore the system [6,7,8].

MCs are derived from myeloid progenitors of the primitive yolk sac and are the resident immune cells of the CNS [9]. They actively seed the CNS parenchyma during mid-embryonic development [10] and colonize the retina through the optic nerve (ON) and the ciliary body (CB) [9]. In the adult retina they are present in the retinal nerve fiber layer (RNFL), ganglion cell layer (GCL), inner plexiform layer (IPL), and outer plexiform layer (OPL) [4,11]. In health, MCs maintain the homeostasis, prune the synapses and patrol the tissue [12]. Generally, they are found in a resting state, a definition that has been recently revisited, and it is now known as a surveying state because MCs do not rest but actively scan their surroundings in search of damage or infection [8,13]. In this state, they have long and widely ramified branches and small cell bodies [14]. They express the classical marker ionized calcium binding adaptor molecule (Iba1) [15] and show a low staining level of CD68.

As a result of an injury or infection, MCs change their morphology to an activated state, losing their ramifications and becoming ameboid [14,16]. When activated, MCs are capable of migrating to the site of damage and phagocytose cellular debris, to maintain the tissue integrity (reviewed in [1,2]). CD68 expression increases as the MC change to a phagocytic state, indicating its activation [17,18]. MCs can undergo two different types of activation: M1 and M2 [19,20,21]. M1 microglia, defined as classical activation, usually acts in the first line of defense and they are responsible of the production of pro-inflammatory mediators with cytotoxic properties, such as interleukin-1β (IL-1β), tumour necrosis factor-α (TNF-α), or reactive oxygen species (ROS) [19] through the activation of the NF-kB signaling [22]. M2 microglia, defined as alternative activation of MCs through STAT1 and JAK1 phosphorylation [22], express neurotrophic factors, such as insulin-like grow factor-1 and anti-inflammatory cytokines, such as tumour growth factor-β (TGF-β), IL-13, IL-10 and IL-4a, facilitating tissue repair and reconstruction of the extracellular matrix [23,24,25]. In response to neuronal apoptosis, M2 microglia overexpress typical M2 markers, such as Arg1, Fizz1 and CD206, and alleviate the damage tissue by phagocytosing apoptotic cells [19]. CD206 is a transmembrane glycoprotein, member of the C-type lectin family highly expressed by macrophage/MCs or dendritic cells in an inflammatory environment [2,17,19,24]. Therefore, MCs after a CNS insult can have both sides of the same condition: detrimental/neurodegenerative or beneficial/neuroprotective functions [20].

Astrocytes derive from the neuroectoderm and participate in the metabolic support of neurons [26,27]. During retinal development, astrocytes migrate from optic disc along the blood vessels. In adult retinas, astrocytes are located in the innermost retinal layer [7,26]. The main functions of astrocytes are to maintain the blood-retina barrier and to provide neurotrophic support, as well as to enhance mechanical support for degenerating neurons (reviewed in [7]). Two phenotypes of astrocytes, A1 and A2, have been described as neurotoxic and neuroprotective astrocytes [28], respectively, in a parallelism with the M1 and M2 microglial/macrophage activation. Neurotoxic A1 astrocytes are recognized through complement 3 (C3) activation [28], although more analysis is needed to establish the differences between the two phenotypes (reviewed in [29]).

Müller cells are the predominant glial cells in the retina, and their body spans the entire retinal thickness, from the RNFL to the outer nuclear layer (ONL) [30,31]. Müller cells are the anatomical and functional link between retinal neurons and blood vessels, together with astrocytes. They perform a plethora of functions, such as maintaining the retinal structure and neuronal function, promoting synapse formation and scavenging neurotransmitters [7].

In their resting state, astrocytes and Müller cells express low-medium levels of type III intermediate filaments, GFAP and vimentin, respectively. After an injury, both become hypertrophic and over-express these cytoskeletal proteins [30,32]. In addition, hypertrophic Müller cells express GFAP, and thus their activation can be easily recognized [32].

Optic nerve crush (ONC) is a very well defined lesion that results in retinal ganglion cell (RGC) degeneration. The temporal course of ONC-induced RGC degeneration in rats and mice has been described in depth by our group [33,34,35,36,37]. In both species there are two phases of RGC loss: the first one occurs up to 9 (mice) or 14 (rats) days post-lesion (dpl), and it is characterized by a fast and lineal RGC degeneration, surviving ~15% of RGCs at these time-points. Thereafter, RGCs’ death continues steadily but slowly [34,35,38]. As a consequence of RGC death, MCs become activated, changing their morphology, migrating and phagocytosing the dead RGCs [11,34]. Macroglial cells also react, changing their shape, size and number, and regulating the immune response. Astrocytes and Müller cells increase the expression of GFAP as a sign of reactivity [27], become hypertrophic and form a scar-border to seal the damaged tissue [29].

Furthermore, these responses to ONC are not limited to the injured retina. Recently, it has been described that lesions do affect not only the injured retina but also the contralateral one—the so-called contralateral effect [39]. As theorised by [40], the uninjured contralateral retina responds as the injured one but in a subdued way showing, depending on the original insult, MC activation [11], phagocytic MC appearance [34], astrocyte and Müller cell activation [27,31,41], and loss of RGCs [42].

The dual role of glial cells has acquired great importance in the understanding of neuro-inflammatory and degenerative diseases. It is not still clear how these phenotypes of microglia and astrocytes work together [21], whether they coexist [20], and whether glial cell beneficial roles contribute to the contralateral effect after an unilateral damage.

Therefore, the main purposes of this study are: (i) to characterise the two states of MCs in intact mouse retinas; (ii) to analyse the temporal course of MC activation and M2 appearance in injured and contralateral retinas after unilateral ONC; and (iii) to analyse astrocyte and Müller cell activation, in both injured and contralateral retinas.

## 2. Results

### 2.1. Microglial Cells

All CD68^+^ cells were Iba1^+^ but not vice versa. In addition, all CD206^+^MCs were CD68^+^ and Iba1^+^. Thus, changes of MC density were assessed with Iba1, whereas activation states were assessed with CD68 and CD206.

#### 2.1.1. Intact Retinas

In intact retinas, Iba1^+^MCs were found in the RNFL, GCL, IPL and OPL with a ramified morphology, indicative of a resting/surveying state (Figure 1A). In the GCL, Iba1^+^MCs were evenly distributed (Figure 1B), as described before [4,11]. Expression of CD68 (Figure 1C,D) and CD206 (Figure 1E,F)—markers of MC activation—was observed in very few MCs that were positive for both markers and preferentially located around the optic nerve (ON) and the ciliary body (CB, Figure 1D,F). Their morphology quite differed from the ramified phenotype of the resting Iba1^+^MCs because of its activation.

#### 2.1.2. Injured Retinas

In retinal cross-sections (Figure 2), morphological activation of MCs and expression of CD68 was clearly observed from day 5 (Figure 2B) compared to intact retinas (Figure 2A). Most CD206^+^MCs were located in the GCL (Figure 1E,F) and their number was very low, thus the analysis of this population was carried out in flat mounted retinas.

In the total retina (inner and outer) there was a progressive and parallel increase of CD68^+^MCs and Iba1^+^MCs that was significantly higher than intact at 3, 5 and 9 dpl and decreased to basal levels from 21 dpl (Figure 2C). When we analyzed the inner and outer retina separately, we observed that MC density increased in the inner retina and decreased in the outer one. These data suggest that MCs migrate from the outer retina to the GCL where RGCs are dying (Figure 2C).

In the GCL captured from flat mounts, MCs showed already at day 1 a clear CD68 expression (Figure 3A,B). MC density significantly increased at 5, 9 and 21 dpl, and their number was back to normal at 45 dpl. However, the density of activated CD68^+^MCs increased significantly at day 3 and remained higher than control up to 45 dpl (Figure 3C).

Topographically, CD206^+^MCs were scattered across the GCL (Figure 4A). There was an inverse relationship between the increase of the total number of CD206^+^MCs across the whole GCL (Figure 4) and the decrease of their number around the ON head (Figure 4B and Figure 5). Interestingly, there was also an increase of the number of CD206^+^MCs in the CB, which preceded that of the retina (Figure 4B and Figure 5). In intact retinas, CD206^+^MCs were rounded, with few or no ramifications. After the injury they became elongated and thinner, and in the ON head and CB showed a polarized orientation (Figure 5).

#### 2.1.3. Contralateral Retinas

MC response in the contralateral un-injured retinas followed the same course as in the injured ones but, as expected, this response was a shadow of the injured one. Thus, in the total retina (inner and outer) and in the inner retina, there was a significant increase of Iba1^+^MCs and activated CD68^+^MCs at 5 dpl, while in the outer retina the density of Iba1^+^MC as well as their activation decreased at all time points (Figure 6A–C).

In the GCL, the density of Iba1^+^MCs increased significantly at 5 days, while the density of activated CD68^+^MCs was significantly higher from 1 to 45 days after ONC (Figure 7).

Contrary to the response in the injured retina, the total number of M2 CD206^+^MCs decreased in the contralateral retina. Around the ON head, CD206^+^MCs were observed clearly activated, changing their morphology in a similar way to those in the injured retina. Their number decreased immediately after ONC in the ON head and GCL, and remained low up to 45 dpl. In the CB, the number of CD206^+^MCs increased significantly at 5 days, decreasing afterwards (Figure 4C and Figure 5B).

### 2.2. Macroglial Cells

#### 2.2.1. Injured Retinas

In intact retinas, expression of GFAP and vimentin was restricted to the GCL (Figure 8A). After axotomy, the GFAP and vimentin signal increased in the GCL at 5 and 9 days and decreased at 21 days, indicating a transitory hypertrophy of astrocytes and Müller cells’ end feet (Figure 8B). Regarding GFAP expression in Müller cells’ stalks and somas (from the inner to the outer retina), it was surprising to see that there was not much activation, except at 9 days after the lesion when GFAP staining was observed along Müller cells. Accordingly, the increase of vimentin signal across the retina stacks was subtle but evident from 5 to 21 days after the lesion.

#### 2.2.2. Contralateral Retinas

In the contralateral retinas, there were no qualitative changes in the expression of these two proteins compared to intact retinas (Figure 9A), except at day 3 post-lesion when in the GCL the signal of GFAP and vimentin was brighter than at the rest of the time points (Figure 9B), indicating a transient astrocyte and Müller cells’ end feet hypertrophy.

## 3. Discussion

We studied the course of glial cell activation after an unilateral optic nerve crush in mice, and show that glia from both injured and contralateral retinas react to the lesion.

This bilateral response, the so-called contralateral effect, is being intensively studied in the visual system, and during the past decade more pieces were added to this puzzle [6,18,27,30,34,40,41,42,43], the physiological mechanisms of which are still unknown (reviewed in [39,41]). Still, in all reports, the characteristics of this effect, postulated by Shenker et al. (2003) [40], prevail: first, the damage must reach a minimal magnitude to elicit a contralateral response, and second, the response of the contralateral region is the same as that of the damaged one but attenuated in time and magnitude. Our data here adhere to those characteristics, as the microglial and macroglial activation observed in the injured retinas is stronger, starts earlier and lasts longer than in the contralateral ones. Perhaps the most controversial results are the long-lasting expression of the phagocytosis marker CD68 in both retinas, and the unexpectedly low activation of Müller cells.

### 3.1. Microglial Response

Microglial cells that have phagocytosed dead RGCs, visualized by transcellular labeling [34,44,45], appear at 3 days after ONC in both retinas. In the injured ones, their number increases linearly as the RGCs die with a parallel but opposed kinetics. In the uninjured retina, RGCs also die but their loss is very slow and does not progress further from 9 or 45 days (depending on the lesion site, see below [39,42]). Consequently, the number of transcellularly labelled MCs remains fairly constant.

Our data here fit with those previous studies [34], adding three important sets of data: increased microglial density, a long-lasting phagocytic activated state and M2 MCs dynamics.

Microglial density increases in the inner retina and GCL, decreasing in the outer retina. With time, MC density returns to normal values in the inner but not in the outer retina. This dynamic behaviour has been already shown in rats [11], and it may indicate either that the population of MCs resident in the GCL is not sufficient to cope with the massive loss of RGCs and/or that the death of RGCs attracts them to the GCL. The latter hypothesis is more likely for the contralateral site.

The MC increase lasts longer and is higher in the injured (1.4–2.15-fold from 3 to 9 days) than in the contralateral (1.6-fold at 5 days) retinas. The time course of increased MCs’ density in the injured retinas fits with the time window in which most RGCs die after axotomy (from 3 to 21 days [34,35,42]). In the contralateral retina, this increase precedes the significant loss of RGCs [42] but not the appearance of transcellularly labelled MCs [34].

Interestingly, although MC density in the GCL returns to baseline over time, the MCs are not in a resting/surveillance state but are activated, as they overexpress CD68. The presence of CD68^+^MCs is explained by the fact that in both retinas there are RGCs dying for a long time after the lesion. In the injured retina, >90% RGCs die during the first 2–3 weeks, and the remaining ones die steadily but very slowly up to 3 months (e.g., between days 45 and 90 post-injury between ~500–800 RGCs die [35,42]). In the contralateral retinas, RGC death is very slow: when the lesion is performed at 0.5 mm from the optic head of the injured retina, transcellularly labelled MCs appear very early [34], but the loss of RGCs reaches significance at 45 days [42]. However, the question arises, are microglial cells activated because of the constant RGC death or are RGCs dying because of the unresolved microglial activation?

Alternative M2 activation refers to MCs that are phagocytic—to help clear the tissue and restore homeostasis, hence their high expression of CD68—and that secrete anti-inflammatory mediators, to dampen the acute inflammatory environment created by the insult. They express M2-specific antigens, such as CD206 [24,26].

To our knowledge, this is the first report describing M2 MCs in the retina. Their morphology and location are quite different from that of resting or classically activated MCs. In intact retinas they are few (~250/per retina) with large, slightly elongated and thick somas, few or no ramifications, and are found mainly in the ciliary body and around the optic-nerve head.

Upon injury, they become thinner and longer, as if they were polarized because in the optic-nerve head they arrange radially along the RGC axons, while in the CB they are perpendicular to the retina. One day after the lesion, M2-MCs decrease around the optic-nerve head and increase in the CB, where they remain significantly higher up to day 21. In the GCL, their number increases 2.3–2.6 times at 5 and 9 days, and they are found throughout it. This increase occurs, again, during the phase of maximum RGC death. Then, they return to normal levels in the CB and GCL, but remain below normal levels in the optic-nerve head. In the contralateral retinas, their morphology does not change in the CB but it does in the optic-nerve head at day 5. Contrary to the dynamics observed in the injured retinas, their number decreases in the GCL and the optic-nerve head, and it is maintained in the CB after a transient increase.

What is the meaning of this behaviour? Do they populate the injured retina through the optic nerve and the CB? Do they migrate from the contralateral retina through the optic chiasm to the injured retina?. And most importantly, what is their function in this model?

In the spinal cord it has been shown, as we show here, that there is a quick and maintained response of classically activated MCs in the lesion site that overwhelms the comparatively smaller and transient M2 response, which promotes a regenerative growth in adult sensory axons [46]. Thus, would a longer and stronger increase of M2 MCs in the retina rescue RGCs, delay their death and/or increase axonal regeneration? Further experiments are needed to elucidate this point.

### 3.2. Macroglial Response

Surprisingly, macroglial activation, although observed in both retinas and more so in the injured ones, was not very strong. It was mostly restricted to the GCL, indicating that astrocytes and Müller cells’ end feet are the main responders to RGC death.

Müller cell hypertrophy assessed by expression of GFAP and overexpression of vimentin in their stalks, somas and distant processes, was very low, and only observed in the injured retina. The end feet of Müller cells form the inner basal lamina, which is essential for RGC survival [37]. Thus, we expected a stronger reaction of Müller cells to RGC death.

Hypertrophy of the entire Müller cells has been observed in both retinas in models of acute ocular hypertension [25]. This is a very different model from axotomy, as it causes degeneration of photoreceptors [38]. Photoreceptor degeneration triggers a strong activation of Müller cells as well as microglial cells, but not so much astrocyte activation [32].

Optic-nerve axotomy is a clean lesion that specifically affects RGCs, and it is not then surprising that astrocytes, connected to RGC axons and vessels, hypertrophy when the RGCs are dying. In addition, the activation of MCs in the GCL may activate astrocytes and the inner region of Müller cells, as there is a crosstalk neuron-micro-macroglia. Therefore, it may be that astrocytes are more sensitive to RGC death and/or axonal degeneration, i.e., affectation of the RGC complex, while the whole Müller cell responds when there is loss of photoreceptors and probably INL neurons.

## 4. Materials and Methods

### 4.1. Animal Handling

For this study, adult pigmented C57Bl/6 female mice, averaged weight of 30 g, from the University of Murcia breeding colony were used. All animals were treated according to the European Union guidelines for Animal Care and Use for Scientific Purpose (Directive 2010/63/EU) and the guidelines from the Association for Research in Vision and Ophthalmology (ARVO) Statement for the Use of Animals in Ophthalmic and Vision Research. All procedures were approved by the Ethical and Animal Studies Committee of the University of Murcia, Spain (A1320140704 approved in July 2014, extended 30 July 2020, valid until 2025).

Before the surgery, animals were anesthetized by intraperitoneal injection of a mixture of xylazine (10 mg/kg; Rompum, Bayer, Kiel, Germany) and ketamine (60 mg/kg; ketolar, Pfizer, Alcobendas, Madrid, Spain). Analgesia was provided by subcutaneous administration of buprenorphine (0.1 mg/kg; Buprex, Buprenorphine 0.3 mg/mL; Schering-Plough, Madrid, Spain). To avoid corneal desiccation, the eyes were covered with Tobrex (Alcon S.A., Barcelona, Spain). Animals were sacrificed with an intraperitoneal injection of an overdose of sodium pentobarbital (Dolethal, Vetoquinol; Especialidades Veterinarias, S.A., Alcobendas, Madrid, Spain).

### 4.2. Experimental Design

In experimental animals, the left ON was intraorbitally crushed at 0.5 mm from the optic disc for 10 s using watchmaker’s forceps, following previously described methods [33,35,42]. All surgeries were carried out by the same experienced investigator. After the surgery, the retinal blood flow of the eye fundus was assessed to confirm the absence of retinal ischemia. Both left injured and right contralateral retinas were analyzed at different times post-lesion (dpl): 1, 3, 5, 9, 21 and 45 dpl (n = 8/group). As control, intact retinas were used (n = 6).

### 4.3. Tissue Preparation

Animals were perfused transcardially with 0.9% saline solution followed by 4% paraformaldehyde (PFA) in 0.1 M phosphate buffer, and eyes were fixed one extra hour in 4% PFA. After that, retinas were dissected as flattened whole-mounts [33], leaving the CB attached (n = 4/group), or prepared for cryostat sectioning of 15 µm thick (n = 3/group), as previously described [35].

### 4.4. Immunohistofluorescence and Antibodies

Immunodetection was carried out in whole-mounted and cross-sectioned retinas as previously reported [34,35]. Primary antibodies are detailed in Table 1.

Secondary detection was carried out with the appropriate combination of Alexa-coupled secondary antibodies each diluted 1:500 (Molecular Probes, Thermo Fisher Scientific, Madrid, Spain): donkey anti-rabbit (488, Cat.A32790), donkey anti-goat (555; Cat.A32816), donkey anti-goat (594; Cat.A11058) and goat anti-rat (647; Cat.A21247).

### 4.5. Image Acquisition

Images of whole-mounted and cross-sectioned retinas were acquired using an epifluorescence microscope (Leica DM4 B, Heidelberg, Germany) equipped with the following filters: dichroic filters for fluorescein (BP 470/40, LP 525/50), rhodamine (BP545/25, LP 605/70), far red (BP 620/60/LP 700/75) and Dapi (BP 350/50, LP 460/50). Images were taken at 20x magnification, reconstructed from 154 individual images (11 × 14) [33] controlled by the software LAS X (Leica Application Suite) and the camera Leica (Microsystems Heidelberg GmBH, Germany). GFAP and vimentin acquisition settings were the same for all conditions, to allow qualitative comparisons.

### 4.6. Quantification and Analysis

MCs were blind analyzed in cross-sections and whole-mounts, using Iba1 for detecting all MCs, CD68 for activated MCs and CD206 for M2-MCs.

In cross-sectioned retinas, mean density of Iba1^+^ and CD68^+^MCs (cells/mm^2^) in the outer retina (OPL to retinal pigmented epithelium) and inner retina (RNFL to IPL) was quantified in 3 representative sections of each animal (n = 3/time-point). For that, outer and inner retinas were manually outlined with ImageJ software (developed by Wayne Rasband, National Institutes of Health, Bethesda, MD, USA; https://imagej.nih.gov/ij, 8 August 2021), and positive cells in each area were manually dotted. Finally, the mean density was calculated. The total MC density was inferred as the sum of both.

In whole-mounted retinas, photographs were taken with the retina upwards, focusing on the GCL. Iba1^+^MC and CD68^+^MC density (cells/mm^2^) was manually quantified in 12 individual images of each sample (n = 4/time-point), 3 from each retinal quadrant (centre, medium and periphery) [33]. CD206^+^MCs were very few and they were distributed across the retina; thus, their total number was manually quantified. As most CD206^+^MCs were observed close to the ON and in retinal periphery, the whole number of CD206^+^MCs in the CB and in a ring centered on the ON of radius 283 µm (0.25 mm^2^) was analyzed separately.

CD206^+^MCs were manually dotted on the photomontages, quantified and their topography represented by neighbour maps that depict the number of neighbours around a given cell in a radius of 0.200 mm with a colour scale that goes from 0–2 neighbours (purple) to >21 neighbours (dark red).

Finally, macroglia cells were qualitatively analyzed in cross-sectioned retinas using GFAP staining for astrocytes and activated Müller cells (n = 3/group).

### 4.7. Statistics

Data are presented as mean ± standard deviation (S.D.) and analyzed with GraphPad Prism v.7 (GraphPad Software Inc., La Jolla, CA, EEUU) using the nonparametric Kruskal–Wallis test. Differences among groups were considered significant when *p* < 0.05.

## 5. Conclusions

Axonal damage caused by trauma or disease results in immediate and irrevocable loss of function in mammals. To date, there are no therapies that work, leaving patients with lasting disability. Animal models are very useful for understanding the whys and hows of neuronal damage and rescue, but until now most research has focused on studying the fate of neurons in injured tissue. We now know that the effect of a unilateral injury extends to the uninjured contralateral area, and that glial cells play a very important role in this bilateral response.

As we showed here, after unilateral optic nerve axotomy, microglial cells from both retinas invade the GCL at the expense of the outer retina, remaining chronically activated, while macroglial activation is transient and limited to the GCL. Thus, axotomy elicits a significant response in time and space, and we wonder whether chronic MC activation is caused by the continued loss of RGCs or whether microglial activation is the cause of the protracted RGC death.

## Figures and Tables

**Figure 1 ijms-22-08517-f001:**
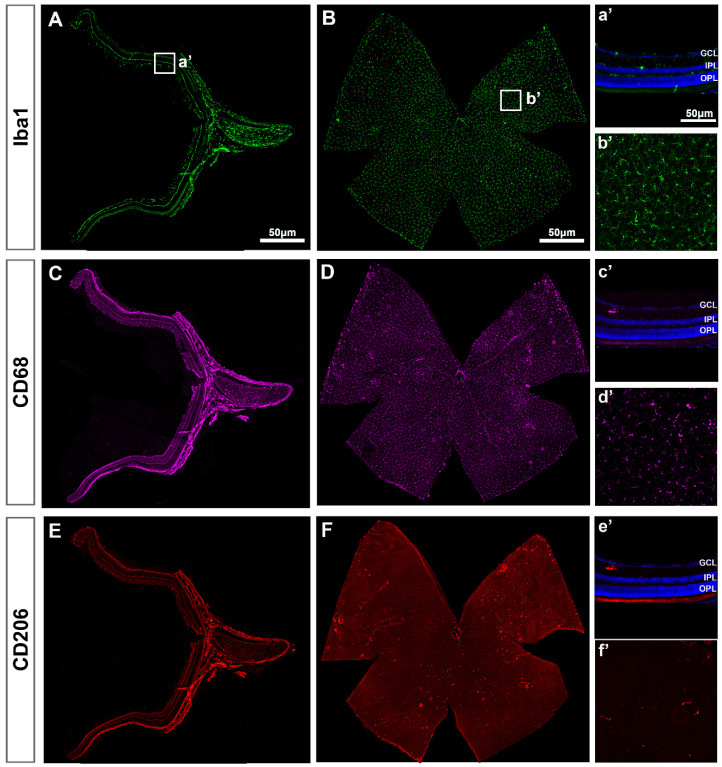
Microglial cells in intact retinas. (**A**–**F**): Photomontages of a representative retinal cross-section (**A**,**C**,**E**) and flat mount (**B**,**D**,**F**) where Iba1, CD68 and CD206 have been immunodetected to identify all MCs (Iba1) and their activated states (CD68 and CD206). Magnifications in (**a’**–**f’**) are from the framed areas in (**A**–**F**). GCL: ganglion cell layer; IPL: inner plexiform layer; OPL: outer plexiform layer.

**Figure 2 ijms-22-08517-f002:**
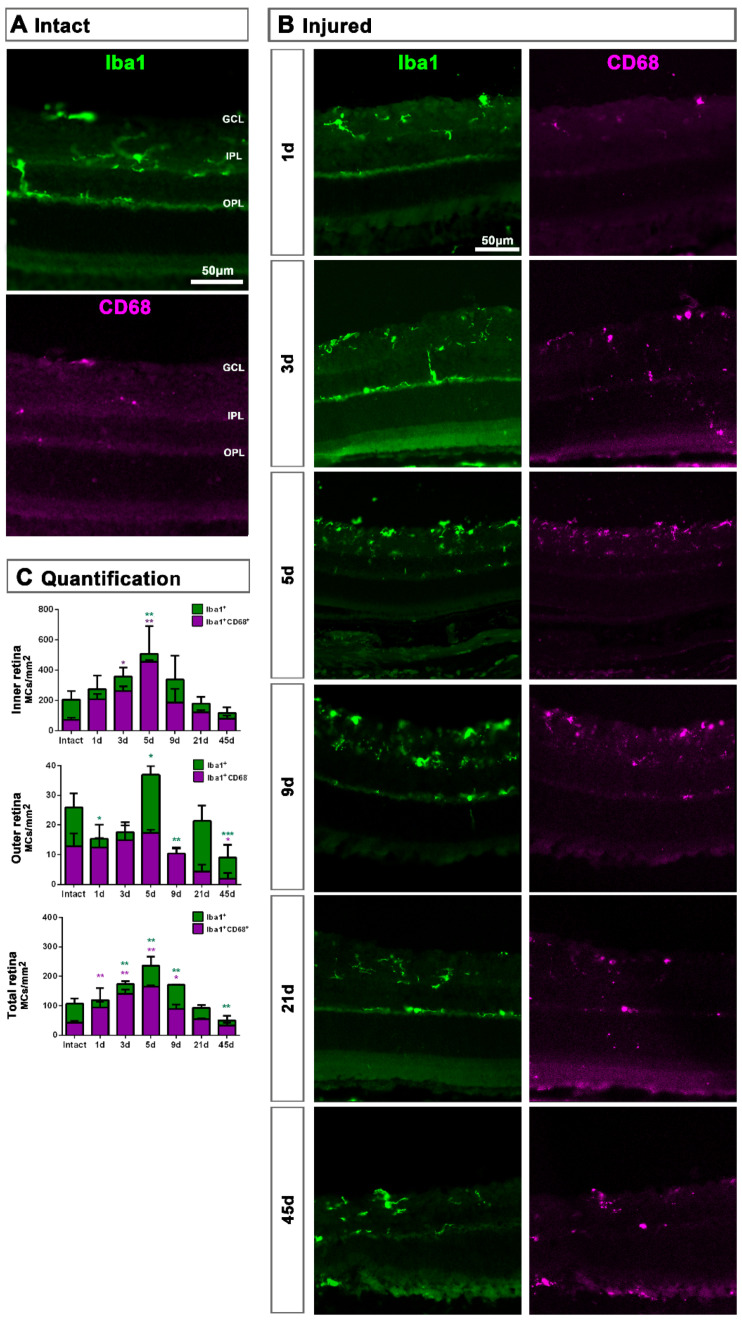
Microglial cell density and activation in axotomized retinas. (**A**,**B**): Magnifications from intact (**A**) and injured (**B**) cross-sectioned retinas analyzed at increasing times post-ONC (**B**) and immunostained against Iba1 and CD68. (**C**). Bar graphs showing the mean ± SD density (MCs/mm^2^) of Iba^+^MCs and CD68^+^MCs in the inner, outer and total retina. Kruskal–Wallis test, significantly different compared to intact retinas, * *p* < 0.05, ** *p* < 0.01, ***p<0.001. GCL: ganglion cell layer, OPL: outer plexiform layer, IPL: inner plexiform layer.

**Figure 3 ijms-22-08517-f003:**
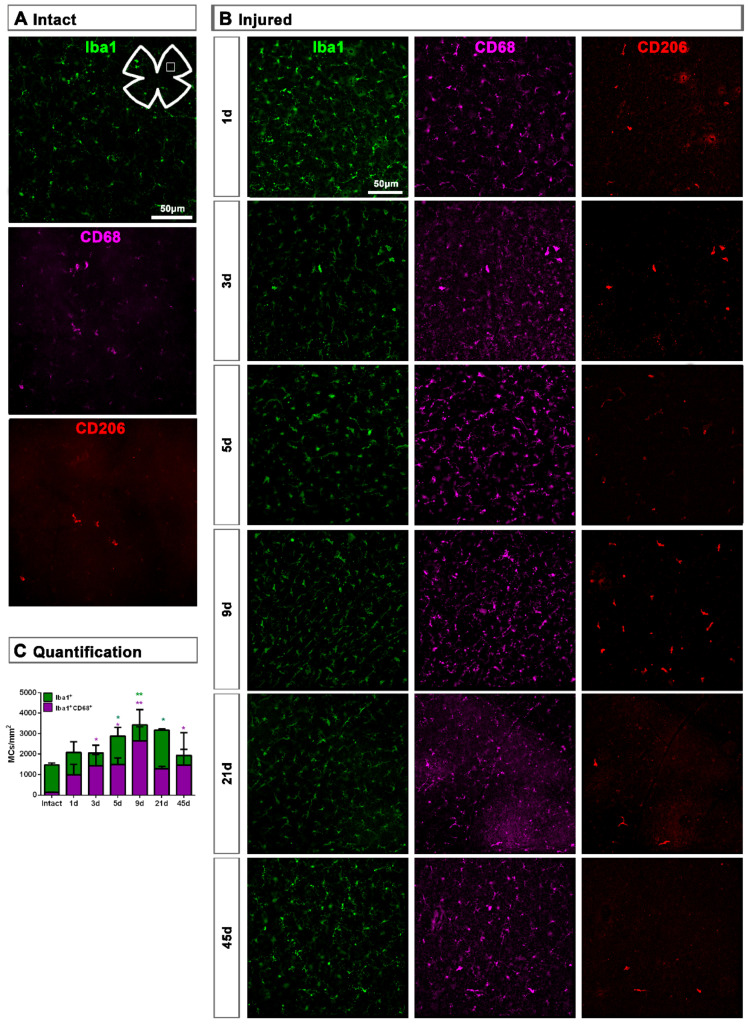
Microglial cell density and activation in the ganglion cell layer of axotomized retinas. (**A**,**B**): Magnifications from intact (**A**) and injured (**B**) flat-mounted retinas focussed on the GCL and showing Iba1, CD68 and CD206 immunostaining. (**C**): bar graphs showing the mean ± SD density (MCs/mm^2^) of Iba^+^MCs and CD68^+^MCs in the GCL. Kruskal–Wallis test, significantly different compared to intact retinas, * *p* < 0.05, ** *p* < 0.01.

**Figure 4 ijms-22-08517-f004:**
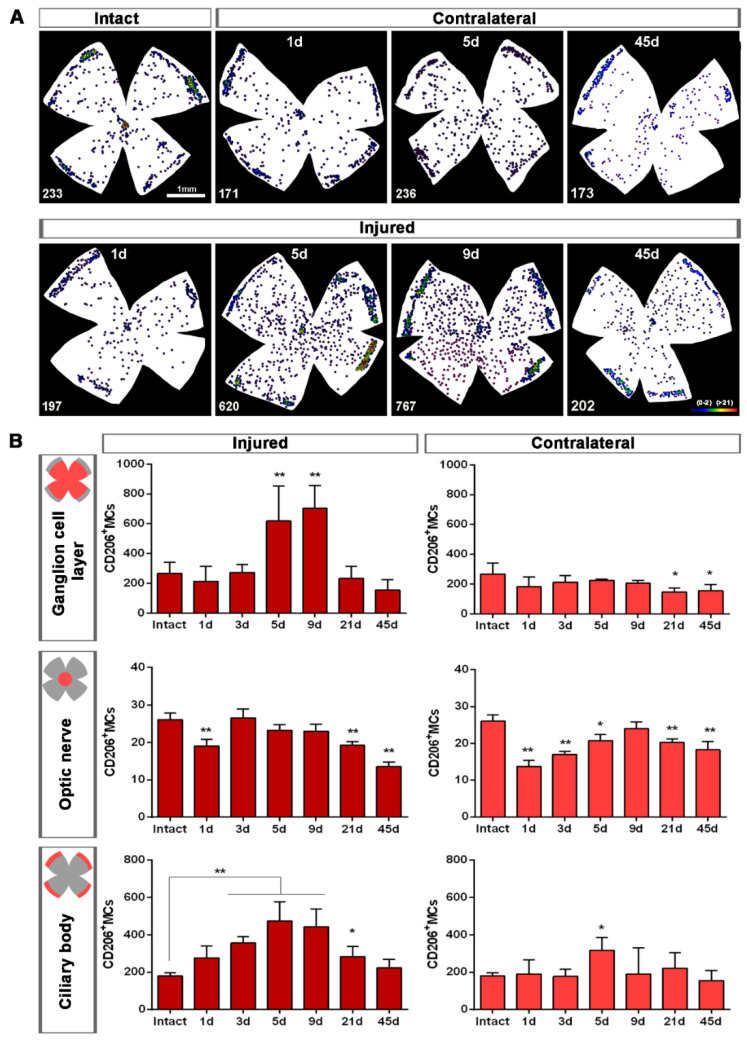
Topographical and quantitative analysis of CD206^+^microglial cells (MCs). (**A**): Neighbour maps showing the retinal distribution of CD206^+^MCs in intact, injured and contralateral retinas. Scale bar in A = 1mm. Below each map is the number of cells counted in the original retina. (**B**): Quantitative analysis of CD206^+^MCs. Bar graphs showing the total number ± SD of CD206^+^MCs in the ganglion cell layer (GCL) excluding the CB (GCL, top row); in the optic-nerve head (ON, middle row) and the ciliary body (CB, bottom), in injured (left) and contralateral (right) retinas at increased times post-ONC (d). Kruskal–Wallis test, significantly different compared to intact retinas, * *p* < 0.05, ** *p* < 0.01.

**Figure 5 ijms-22-08517-f005:**
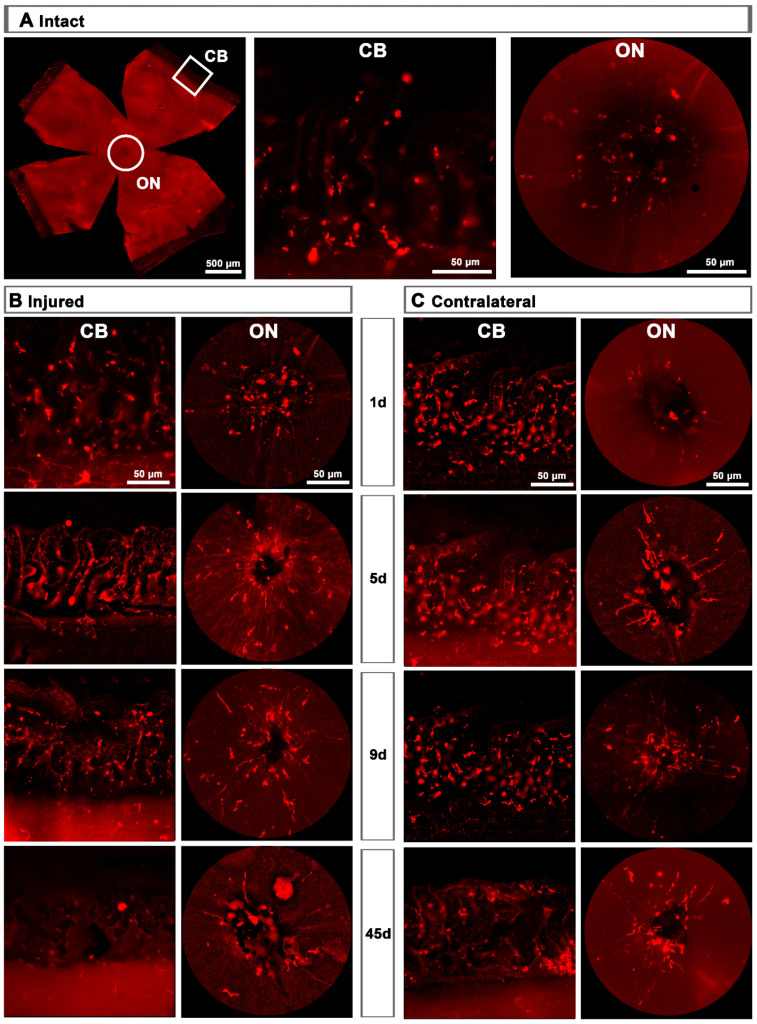
CD206^+^ microglial cells (MCs) in the optic-nerve head and ciliary body. (**A**): Left: photomontage of an intact flat-mounted retina showing CD206 immunodetection. Middle, magnification of the ciliary body (CB). Right, magnification of the optic-nerve head (ON). These magnifications are from the framed areas shown in the photomontage. (**B**,**C**): CD206^+^MCs in the CB and ON head of injured (**B**) and contralateral (**C**) retinas at increasing times post-lesion (d).

**Figure 6 ijms-22-08517-f006:**
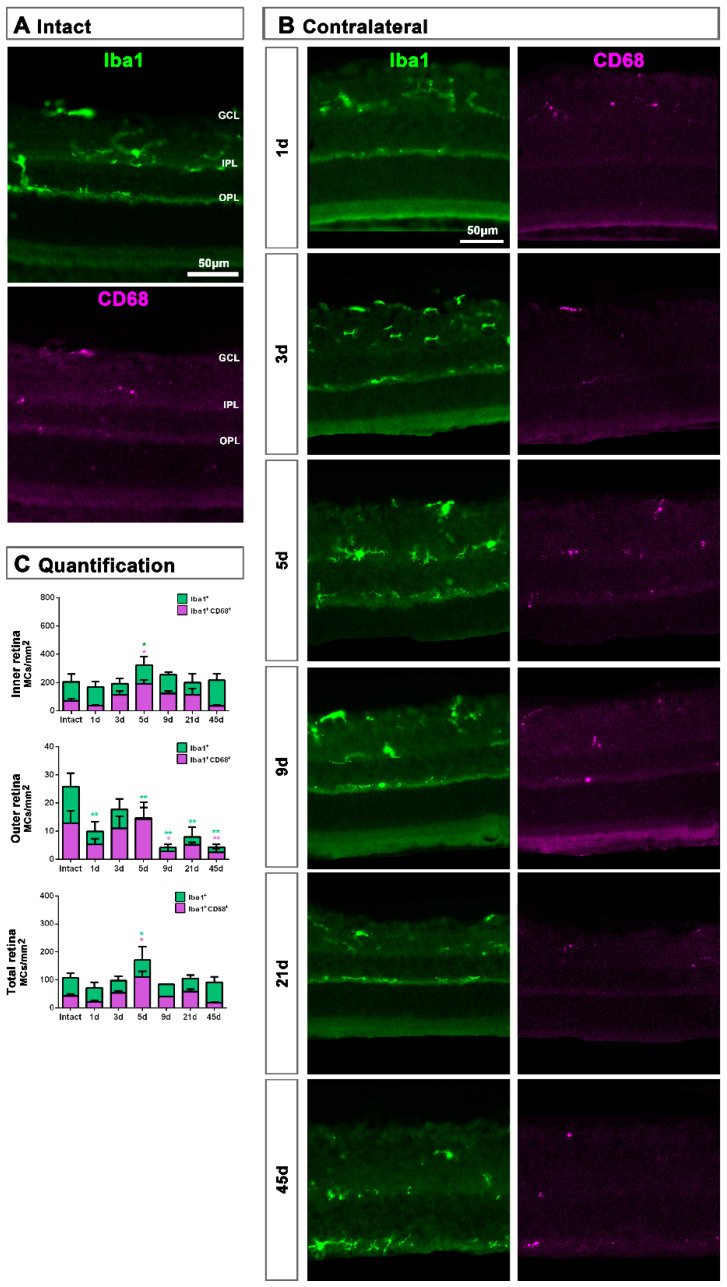
Microglial cell density and activation in contralateral uninjured retinas. (**A**,**B**): Magnifications from intact (**A**) and contralateral uninjured (**B**) cross-sectioned retinas analyzed at increasing times post-ONC (**B**) and immunostained against Iba1 and CD68. (**C**): Bar graphs showing the mean±SD density (MCs/mm^2^) of Iba^+^MCs and CD68^+^MCs in the inner, outer and total retina. One-way ANOVA test, significantly different compared to intact retinas, * *p* < 0.05, ** *p* < 0.01. GCL: ganglion cell layer, OPL: outer plexiform layer, IPL: inner plexiform layer.

**Figure 7 ijms-22-08517-f007:**
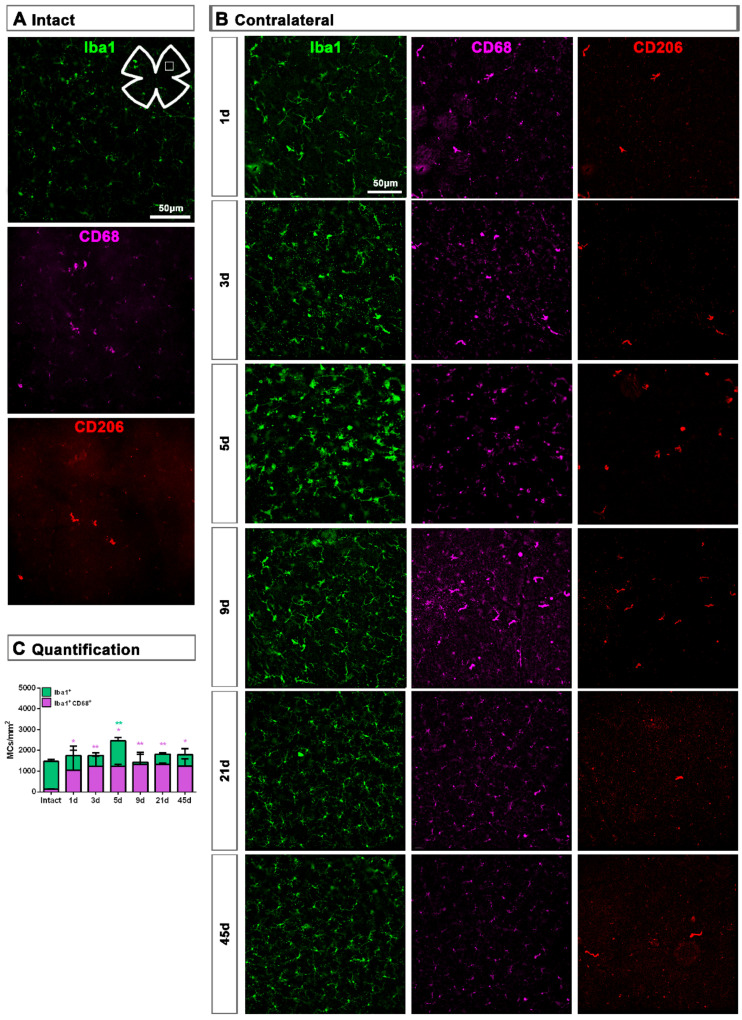
Microglial cell density and activation in the ganglion cell layer of contralateral uninjured retinas. (**A**,**B**): Magnifications from intact (**A**) and contralateral uninjured (**B**) flat-mounted retinas focused on the GCL and showing Iba1, CD68 and CD206 immunostaining. (**C**): Bar graphs showing the mean ± SD density (MCs/mm^2^) of Iba^+^MCs and CD68^+^MCs in the GCL. Kruskal–Wallis test, significantly different compared to intact retinas, * *p* < 0.05, ** *p* < 0.01.

**Figure 8 ijms-22-08517-f008:**
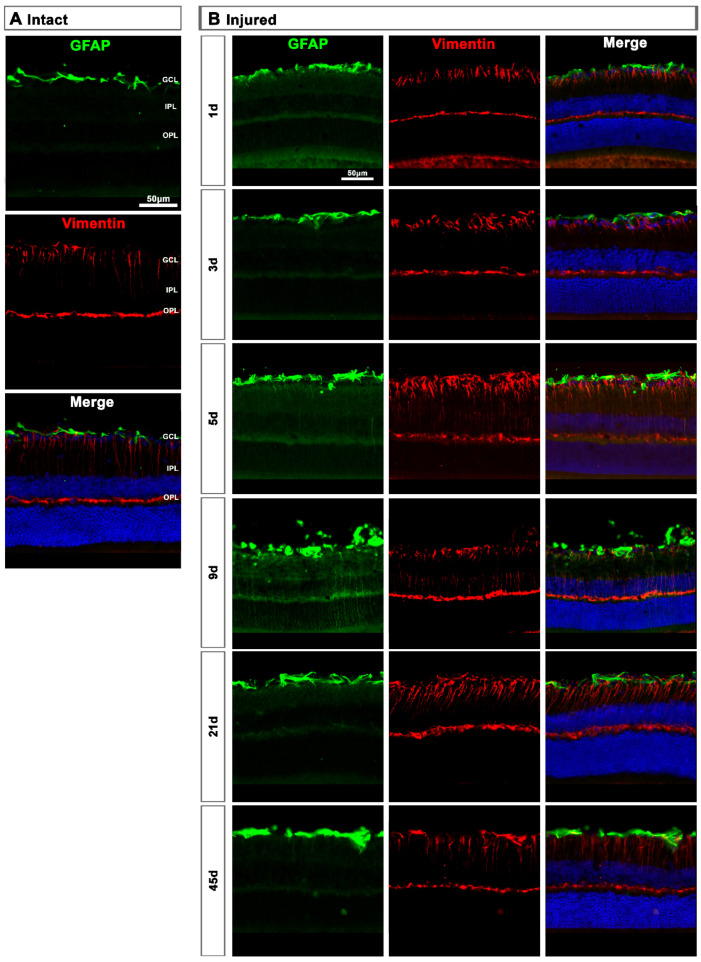
Astrocyte and Müller cell activation in injured retinas. (**A**,**B**): Magnifications from intact (**A**) and injured (**B**) cross-sectioned retinas immunostained against GFAP and vimentin, and counterstained with DAPI. GCL: ganglion cell layer; OPL: outer plexiform layer; IPL: inner plexiform layer.

**Figure 9 ijms-22-08517-f009:**
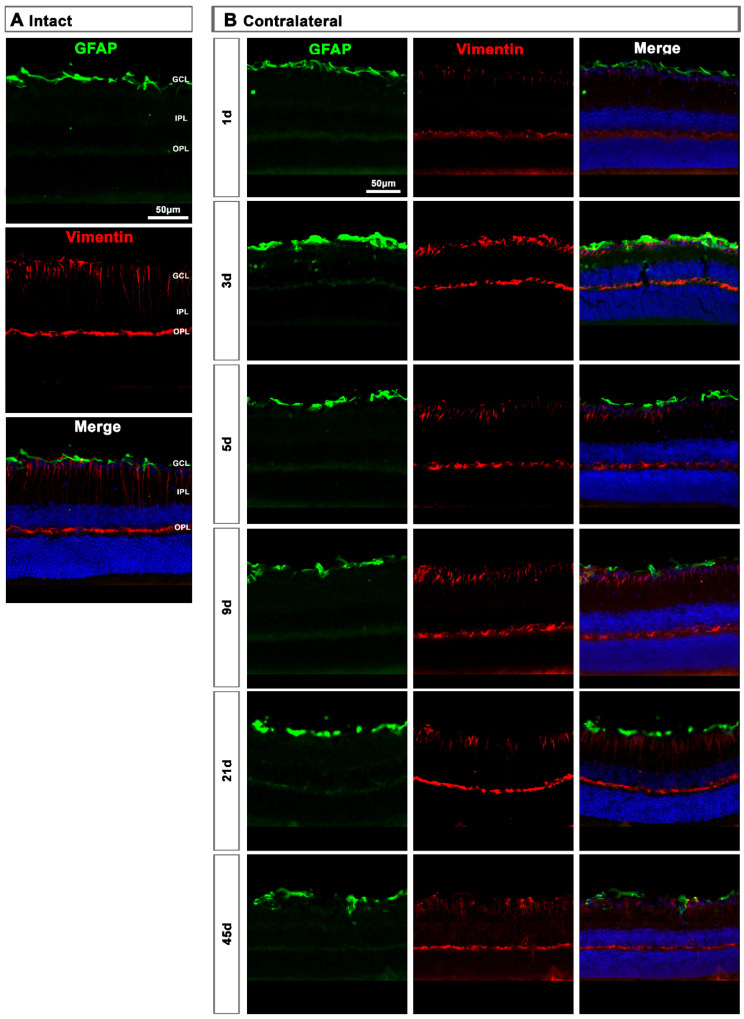
Astrocyte and Müller cell activation in contralateral retinas. (**A**,**B**): Magnifications from intact (**A**) and contralateral uninjured (**B**) cross-sectioned retinas immunostained against GFAP and vimentin, and counterstained with DAPI. GCL: ganglion cell layer; OPL: outer plexiform layer; IPL: inner plexiform layer.

**Table 1 ijms-22-08517-t001:** List of primary antibodies for immunofluorescence staining of retinas.

Cell Type	Antigen	Species	Type	Dilution	Company	Cat. Number
Total MCs	Iba1	Rabbit	Polyclonal IgG	1:500	Abcam	Ab178846
M2 MCs	CD206	Goat	Polyclonal IgG	1:1200	Biotechne	AF2535
Activated MCs	CD68	Rat	Monoclonal IgG2a	1:750	Abcam	Ab53444
Astrocytes	GFAP	Rabbit	Polyclonal IgG	1:500	Sigma-Aldrich	G9269
		Goat	Polyclonal IgG	1:500	Abcam	Ab53554
Müller cells	Vimentin	Goat	Polyclonal IgG	1:250	Santa Cruz	Sc-7557

## Data Availability

Not applicable.

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
