# Peer review of "Axonal Injuries Cast Long Shadows: Long Term Glial Activation in Injured and Contralateral Retinas after Unilateral Axotomy"

_ijms, 2021, doi:10.3390/ijms22168517_

Round 1

Reviewer 1 Report

Although relatively interesting, the present work of  González-Riquelme and colleagues has many serious methodological errors which makes the paper not possible to publish.

  1. The major problem is that the authors did not test the specificity of antibodies used.
  2. Line 322 - the issue how to establish the minimal number of animals to test to get statistically significant data is really disputable. Although, there are some directions provided by Universities or Journals (https://www.nature.com/articles/laban0508-193a) most of researchers agree that depending on kind of experiment the minimal number of experimental animals should be 5 or 6 (I personally agree with this). Thus, n=3 (WM) or n=4 (cryostat) are just too small.
  3. In table 1 the primary type of antibodies used (monoclonal or polyclonal) should be indicated
  4. Line 333 – the authors should provide details (wavelengths) of fluorescent filters used
  5. Line 363 – The authors did not mention what was the post-hoc test used with ANOVA?
  6. The work contains minor spelling errors – in Line 327 “species” instead of “specie” for example

Author Response

DETAILED RESPONSE TO REVIEW

Manuscript ID: ijms-1309684

Title: Axonal injuries cast long shadows: long term glial activation in injured and contralateral retinas after unilateral axotomy

The authors appreciate the work of the editor and reviewers. All comments and criticisms have been very helpful and had greatly improved our manuscript. In the following paragraphs, we answer the specific points addressed by the reviewers. In most instances, we have responded to their suggestions by making changes in the text. For your convenience, we have written in red all the changes made in the manuscript. 

Reviewer #1: 

Although relatively interesting, the present work of González-Riquelme and colleagues has many serious methodological errors which makes the paper not possible to publish.

  1. The major problem is that the authors did not test the specificity of antibodies used.

We thank the reviewer this comment. Regarding the specificity of the antibodies, in this study all antibodies used have been widely tested in previous studies: Iba1 has been described as a marker of all microglial cells [1], CD68 is a marker of activated microglial cells [2-4] and CD206 is the marker of choice for M2 microglial cells [4, 5].

  1. Line 322 - the issue how to establish the minimal number of animals to test to get statistically significant data is really disputable. Although, there are some directions provided by Universities or Journals (https://www.nature.com/articles/laban0508-193a) most of researchers agree that depending on kind of experiment the minimal number of experimental animals should be 5 or 6 (I personally agree with this). Thus, n=3 (WM) or n=4 (cryostat) are just too small.

The authors thank the reviewer for his/her comment. The experimental model optic nerve crush is a very well-established model of retinal ganglion cell degeneration. Previous studies from our laboratory and other groups have shown that this model has a high rate of reproducibility [6-10]. So, in line with the principles of the 3Rs, we have used the minimum number of animals needed to get significance. For retinal cross-sections, 3 different sections of each animal (n= 3 retinas per group and time point) were quantified. For whole-mounted retinas, we used n=4-6 retinas per group and time point. Statistical analysis with these data were robust. and we consider that the samples included in this study are enough to confirm the results.

  1. In table 1 the primary type of antibodies used (monoclonal or polyclonal) should be indicated

The type of antibodies has been included in table 1.

  1. Line 333 – the authors should provide details (wavelengths) of fluorescent filters used

Details of fluorescent filters have been included in methods.

  1. Line 363 – The authors did not mention what was the post-hoc test used with ANOVA?

Statistical analysis has been repeated according to comments of reviewer 3 (nonparametric ANOVA; post hoc Kruskal-Wallis). Graphs have been updated accordingly.

 The work contains minor spelling errors – in Line 327 “species” instead of “specie” for example

We thank the reviewer for his/her suggestions. The manuscript has been thoroughly revised to avoid grammatical, typographical or language errors.

Reviewer 2 Report

 María José González-Riquelme and co-authors describe interesting findings of microglial dynamics after axonal injury with changes also noted in contralateral eyes. A significant and long lasting increase of CD68+MCs is interesting. Additionally, CD206+MCs were found in the ciliary body and around the optic nerve head. Surprisingly macroglia activation seems very limited in the model. Unfortunately authors did not present data on RGC death to correlate these microglial dynamics to neuronal death which would have improved the quality. The following concerns need attention:

  1. It has always been reported that activated MCs are rounded and normal ones are ramified, line 161 states that CD206+ M2 MCs are rounded. Why?
  2. Fig 7 GFAP immunostaining seem to have peaked on 9th day and resolved thereafter without any intervention, does this mean after an axotomy, neuroinflammation subsided by itself after 21 days? Or the extensive activation of Astrocytes suppress muller glia?
  3. Be consistent with stylistic description throughout the manuscript (eg. Fig 1A vs figure 1B).
  4. Could authors provide DAPI in the Figure 1C’, e’ to distinguish layers.
  5. Are all studies performed by a single person or multiple persons? Are the groups blinded to the investigator? If not, please justify.
  6. It is unclear if the increase in M1 MCs resulted in total decreased M2 MCs?

Author Response

DETAILED RESPONSE TO REVIEW

Manuscript ID: ijms-1309684

Title: Axonal injuries cast long shadows: long term glial activation in injured and contralateral retinas after unilateral axotomy

The authors appreciate the work of the editor and reviewers. All comments and criticisms have been very helpful and had greatly improved our manuscript. In the following paragraphs, we answer the specific points addressed by the reviewers. In most instances, we have responded to their suggestions by making changes in the text. For your convenience, we have written in red all the changes made in the manuscript. 

Reviewer #1: 

Although relatively interesting, the present work of González-Riquelme and colleagues has many serious methodological errors which makes the paper not possible to publish.

  1. The major problem is that the authors did not test the specificity of antibodies used.

We thank the reviewer this comment. Regarding the specificity of the antibodies, in this study all antibodies used have been widely tested in previous studies: Iba1 has been described as a marker of all microglial cells [1], CD68 is a marker of activated microglial cells [2-4] and CD206 is the marker of choice for M2 microglial cells [4, 5].

  1. Line 322 - the issue how to establish the minimal number of animals to test to get statistically significant data is really disputable. Although, there are some directions provided by Universities or Journals (https://www.nature.com/articles/laban0508-193a) most of researchers agree that depending on kind of experiment the minimal number of experimental animals should be 5 or 6 (I personally agree with this). Thus, n=3 (WM) or n=4 (cryostat) are just too small.

The authors thank the reviewer for his/her comment. The experimental model optic nerve crush is a very well-established model of retinal ganglion cell degeneration. Previous studies from our laboratory and other groups have shown that this model has a high rate of reproducibility [6-10]. So, in line with the principles of the 3Rs, we have used the minimum number of animals needed to get significance. For retinal cross-sections, 3 different sections of each animal (n= 3 retinas per group and time point) were quantified. For whole-mounted retinas, we used n=4-6 retinas per group and time point. Statistical analysis with these data were robust. and we consider that the samples included in this study are enough to confirm the results.

  1. In table 1 the primary type of antibodies used (monoclonal or polyclonal) should be indicated

The type of antibodies has been included in table 1.

  1. Line 333 – the authors should provide details (wavelengths) of fluorescent filters used

Details of fluorescent filters have been included in methods.

  1. Line 363 – The authors did not mention what was the post-hoc test used with ANOVA?

Statistical analysis has been repeated according to comments of reviewer 3 (nonparametric ANOVA; post hoc Kruskal-Wallis). Graphs have been updated accordingly.

 The work contains minor spelling errors – in Line 327 “species” instead of “specie” for example

We thank the reviewer for his/her suggestions. The manuscript has been thoroughly revised to avoid grammatical, typographical or language errors.

Reviewer #2: 

María José González-Riquelme and co-authors describe interesting findings of microglial dynamics after axonal injury with changes also noted in contralateral eyes. A significant and long lasting increase of CD68+MCs is interesting. Additionally, CD206+MCs were found in the ciliary body and around the optic nerve head. Surprisingly macroglia activation seems very limited in the model. Unfortunately authors did not present data on RGC death to correlate these microglial dynamics to neuronal death which would have improved the quality. The following concerns need attention:

  1. It has always been reported that activated MCs are rounded and normal ones are ramified, line 161 states that CD206+ M2 MCs are rounded. Why?

This is a very good question. CD206+M2 microglial cells express high levels of CD68 and therefore they are in an activated M2 state.

  1. Fig 7 GFAP immunostaining seem to have peaked on 9th day and resolved thereafter without any intervention, does this mean after an axotomy, neuroinflammation subsided by itself after 21 days? Or the extensive activation of Astrocytes suppress muller glia?

We thank the reviewer for the comment. Indeed, we were surprised as well, as we expected a stronger Müller cell response after. We do not know the underlying causes, but as we pointed out in the discussion, Müller cell hypertrophy may depend on the type of injury and the layer of the retina compromised (lines 276-281).

  1. Be consistent with stylistic description throughout the manuscript (eg. Fig 1A vs figure 1B).

Thank you, we have homogenized the style.

  1. Could authors provide DAPI in the Figure 1C’, e’ to distinguish layers.

The authors thank the reviewer for his/her suggestion. DAPI has been included in cross-sectioned retinas of Figure 1 and the figure has significantly improved.

  1. Are all studies performed by a single person or multiple persons? Are the groups blinded to the investigator? If not, please justify.

Optic nerve crush surgery was always carried out by the same person to avoid variability. Data were analyzed by two different investigators blinded to the groups. This has been included in methods (lines 302 and334, respectively).

  1. It is unclear if the increase in M1 MCs resulted in total decreased M2 MCs?

The results of this study show that M2 MCs are a minority of all MC population. In the injured retina, both M2 and M1 MCs follow the same course of activation. Discussed in line 226.

Reviewer #3: 

The current study is an interesting contribution to molecular mechanisms associated with optic nerve crush in animal models. The methodology is sound, and the results merit publication in IJMS. However, the below concerns need to be addressed by the authors:

  1. The authors have used parametric tests: ANOVA and t-test, although the data has most probably a skewed distribution. The authors could replace these tests with nonparametric ones: Kruskal-Wallis.

We thank the reviewer for his/her comments and the authors agree that a non-parametric test is more appropriate. The statistical analysis has been repeated using non-parametric ANOVA followed by Kruskal-Wallis post-hoc test and the results have been updated.

  1. The plagiarism check using the Turnitin tool revealed some slight elevation in plagiarism (24%). The authors need to do some minor rephrasing to become less than 20%.

We checked the percentage of plagiarism using Turnitin and agree that the total percentage is slightly high.

We have analyzed the report of Turnitin, and observed that the highest percentages, 5% and 3%, correspond to affiliation and author´s name, respectively.

Animal handling paragraph gives a high plagiarism index. However, this is not so, as these sentences are standardized.

The rest of sentences, highlighted with “less than 1% of plagiarism (two or three words)”, are technical terms commonly used in this area of research. such as “microglial activation” or “ganglion cell layer”.

When we exclude this last option (below 1%) from Turnitin, the plagiarism from the original manuscript is 9%.

Even so, the manuscript has been thoroughly revised to avoid plagiarism. In the revised version, Turnitin shows 16% of plagiarism including “<1% plagiarism” and 6% excluding it.

  1. It is known that unilateral optic nerve lesions induce a bilateral response that causes an inflammatory and microglial response in the contralateral un-injured retinas. Evidence is also suggesting that contralateral response involves RGC loss. The reviewer would like to know why the authors did not perform protein assays to understand the inflammatory response and quantify the RGC loss.

We thank the reviewer for this suggestion. The time course of RGC degeneration after optic nerve crush in the injured retina has been described in depth by our group, both in rats [6, 7, 11] and mice [8-10, 12]. Recently, we have also demonstrated that in the contralateral retina, there is also RGC loss at longer times after optic nerve crush [10, 13]. As described above, this model is very reliable due to its low variability.

To avoid the misuse of animals and to comply with the 3 Rs, instead of using more animals to analyze RGC degeneration as well, we have compared these results with our previous studies that describe the time course of RGC degeneration in the same species and after ONC.

Regarding protein assays, this is a very good suggestion, and in fact we are planning further experiments to analyze the molecular mechanisms implicated in the microglial dynamics that has been described in this manuscript.

  1. Lines 40-48, 55-58, and many others throughout the manuscript lack reference (s). The authors are encouraged to add reference(s) at the end of every statement (outside the results section).

The authors thank to the reviewer for his/her recommendation- The manuscript has been revised and references have been added throughout the whole manuscript.

  1. Line 331: Table 2 lists the secondary antibodies, not the primary; please correct.

The name of table 2 has been changed.

  1. Tables 1 and 2 would fit better in the supplementary material.

According to reviewer 1’s comments, it is important to describe which primary antibodies have been used and their specificity, thus we have maintained table 1 in the manuscript. Table 2 has been replaced by a paragraph in the text.

References:

  1. Ito, D.; Imai, Y.; Ohsawa, K.; Nakajima, K.; Fukuuchi, Y.; Kohsaka, S., Microglia-specific localisation of a novel calcium binding protein, Iba1. Brain Res Mol Brain Res 1998, 57, (1), 1-9.
  2. Rojas, B.; Gallego, B. I.; Ramirez, A. I.; Salazar, J. J.; de Hoz, R.; Valiente-Soriano, F. J.; Aviles-Trigueros, M.; Villegas-Perez, M. P.; Vidal-Sanz, M.; Trivino, A.; Ramirez, J. M., Microglia in mouse retina contralateral to experimental glaucoma exhibit multiple signs of activation in all retinal layers. J Neuroinflammation 2014, 11, 133.
  3. Chen, L.; Yang, P.; Kijlstra, A., Distribution, markers, and functions of retinal microglia. Ocul Immunol Inflamm 2002, 10, (1), 27-39.
  4. Chhor, V.; Le Charpentier, T.; Lebon, S.; Ore, M. V.; Celador, I. L.; Josserand, J.; Degos, V.; Jacotot, E.; Hagberg, H.; Savman, K.; Mallard, C.; Gressens, P.; Fleiss, B., Characterization of phenotype markers and neuronotoxic potential of polarised primary microglia in vitro. Brain Behav Immun 2013, 32, 70-85.
  5. Horie, S.; Robbie, S. J.; Liu, J.; Wu, W. K.; Ali, R. R.; Bainbridge, J. W.; Nicholson, L. B.; Mochizuki, M.; Dick, A. D.; Copland, D. A., CD200R signaling inhibits pro-angiogenic gene expression by macrophages and suppresses choroidal neovascularization. Sci Rep 2013, 3, 3072.
  6. Villegas-Perez, M. P.; Vidal-Sanz, M.; Rasminsky, M.; Bray, G. M.; Aguayo, A. J., Rapid and protracted phases of retinal ganglion cell loss follow axotomy in the optic nerve of adult rats. J Neurobiol 1993, 24, (1), 23-36.
  7. Nadal-Nicolas, F. M.; Jimenez-Lopez, M.; Sobrado-Calvo, P.; Nieto-Lopez, L.; Canovas-Martinez, I.; Salinas-Navarro, M.; Vidal-Sanz, M.; Agudo, M., Brn3a as a marker of retinal ganglion cells: qualitative and quantitative time course studies in naive and optic nerve-injured retinas. Invest Ophthalmol Vis Sci 2009, 50, (8), 3860-8.
  8. Galindo-Romero, C.; Aviles-Trigueros, M.; Jimenez-Lopez, M.; Valiente-Soriano, F. J.; Salinas-Navarro, M.; Nadal-Nicolas, F.; Villegas-Perez, M. P.; Vidal-Sanz, M.; Agudo-Barriuso, M., Axotomy-induced retinal ganglion cell death in adult mice: quantitative and topographic time course analyses. Exp Eye Res 2011, 92, (5), 377-87.
  9. Sanchez-Migallon, M. C.; Valiente-Soriano, F. J.; Salinas-Navarro, M.; Nadal-Nicolas, F. M.; Jimenez-Lopez, M.; Vidal-Sanz, M.; Agudo-Barriuso, M., Nerve fibre layer degeneration and retinal ganglion cell loss long term after optic nerve crush or transection in adult mice. Exp Eye Res 2018, 170, 40-50.
  10. Lucas-Ruiz, F.; Galindo-Romero, C.; Rodriguez-Ramirez, K. T.; Vidal-Sanz, M.; Agudo-Barriuso, M., Neuronal Death in the Contralateral Un-Injured Retina after Unilateral Axotomy: Role of Microglial Cells. Int J Mol Sci 2019, 20, (22).
  11. Vidal-Sanz, M.; Galindo-Romero, C.; Valiente-Soriano, F. J.; Nadal-Nicolas, F. M.; Ortin-Martinez, A.; Rovere, G.; Salinas-Navarro, M.; Lucas-Ruiz, F.; Sanchez-Migallon, M. C.; Sobrado-Calvo, P.; Aviles-Trigueros, M.; Villegas-Perez, M. P.; Agudo-Barriuso, M., Shared and Differential Retinal Responses against Optic Nerve Injury and Ocular Hypertension. Front Neurosci 2017, 11, 235.
  12. Sanchez-Migallon, M. C.; Valiente-Soriano, F. J.; Nadal-Nicolas, F. M.; Di Pierdomenico, J.; Vidal-Sanz, M.; Agudo-Barriuso, M., Survival of melanopsin expressing retinal ganglion cells long term after optic nerve trauma in mice. Exp Eye Res 2018, 174, 93-97.
  13. Lucas-Ruiz, F.; Galindo-Romero, C.; Albaladejo-Garcia, V.; Vidal-Sanz, M.; Agudo-Barriuso, M., Mechanisms implicated in the contralateral effect in the central nervous system after unilateral injury: focus on the visual system. Neural Regen Res 2021, 16, (11), 2125-2131.

Reviewer 3 Report

The current study is an interesting contribution to molecular mechanisms associated with optic nerve crush in animal models. The methodology is sound, and the results merit publication in IJMS. However, the below concerns need to be addressed by the authors: 

The authors have used parametric tests: ANOVA and t-test, although the data has most probably a skewed distribution. The authors could replace these tests with nonparametric ones: Kruskal-Wallis.

The plagiarism check using the Turnitin tool revealed some slight elevation in plagiarism (24%). The authors need to do some minor rephrasing to become less than 20%

It is known that unilateral optic nerve lesions induce a bilateral response that causes an inflammatory and microglial response in the contralateral un-injured retinas. Evidence is also suggesting that contralateral response involves RGC loss. The reviewer would like to know why the authors did not perform protein assays to understand the inflammatory response and quantify the RGC loss..   

Lines 40-48, 55-58, and many others throughout the manuscript lack reference (s). The authors are encouraged to add reference(s) at the end of every statement (outside the results section).

Line 331: Table 2 lists the secondary antibodies, not the primary; please correct. 

Tables 1 and 2 would fit better in the supplementary material.  

Author Response

(The authors gave the same response as above.)

Round 2

Reviewer 1 Report

The explanations of the authors are resonable.